# Solid Particle Erosion Behavior on the Outer Surface of Basalt/Epoxy Composite Pipes Produced by the Filament Winding Technique

**DOI:** 10.3390/polym15020319

**Published:** 2023-01-08

**Authors:** Seyit Mehmet Demet, Harun Sepetcioglu, Mehmet Bagci

**Affiliations:** 1Mechanical Engineering Department, Engineering and Natural Science Faculty, Konya Technical University, Konya 42250, Turkey; 2Department of Metallurgy and Materials Engineering, Faculty of Technology, Selçuk University, Konya 42250, Turkey

**Keywords:** erosive effect, BFR/EP pipe, filament winding, erosion resistance

## Abstract

Basalt/epoxy composite pipes in a [±55]_4_ winding configuration were produced on CNC filament winding machines (10 N fiber tension and ~11 mm bandwidth). In the experiments, a 34 m/s impact velocity was set using the double-disc method, and five different particle impingement angles (30, 45, 60, 75, and 90°) were used to determine the erosive effect on the outer surfaces of filament wound composite pipes under the influence of 600 μm erodent particles with angular geometry in the test set, complying with the ASTM G76-95 standard. The winding patterns in the lamina (±55 angle-ply laminate region) and zigzag (±55 zigzag region) regions of BFR/EP pipes were determined to have significant effects on solid particle erosion resistance, as evidenced by the SEM images.

## 1. Introduction

Composites are a class of materials formed by combining components with one or more desired properties in macro- and/or micro-dimensions in accordance with the design purpose (thermal, mechanical, physical, operational, etc.) and in which the desired properties predominate while undesirable properties are eliminated. In recent years, composites have begun to be used in an equal proportion to metals in terms of volume because of their high strength, low weight, easy workability, high chemical-corrosion resistance, high thermal-electrical properties, and superior mechanical properties [1]. Due to the advantages they offer, composite materials are also widely studied in the literature. In some of these applications, the tribological properties of bearings made of carbon- and bronze-reinforced composite material were investigated. In some studies, the tribological properties of various fiber-reinforced composite materials are investigated [2,3]. These studies can be performed in various sets of experiments on tribology. It is seen that some studies conduct research on widely used experimental sets such as pin on disc, while others conduct research on specially designed experimental sets to investigate the application. When the studies investigating erosion wear are examined, there are studies where test sets compatible with the ASTM G76-95 standard are used and the erosive wear behaviors of various fiber-reinforced composites are investigated, as in this study [4]. Reinforcing materials have an important place in the creation of composites, and basalt technology is a pioneer in the development of higher-quality composite materials needed in the sector, focusing on lightness and durability, especially in defense-related applications. Basalt fibers, which can be used in applications between approximately −230 and +700 °C, have a wider application temperature range than E-glass, which can be used between −60 and +450 °C, but do not enter into a toxic reaction with air and water. Basalt fiber reinforcement (BFR) has become increasingly important in industrial applications due to its high chemical and thermal stability as well as its temperature resistance and electrical and sound insulation properties. It also shows good mechanical properties compared to its efficiency-oriented alternatives in the field of use [4,5,6,7,8,9]. Due to these advantages, basalt fiber-reinforced composites are used for insulation purposes in some applications [6], and there are studies on their use in automotive components as well [7]. In another study, the slurry erosion behavior of basalt-reinforced composite material was investigated, and it was stated that thanks to basalt reinforcement, it increased the hardness of the composite material and provided erosion resistance [8].

Al_2_O_3_ (alumina) erodent particles were used to test materials that are found in a wide range of areas, including aerospace applications, energy conversion systems, jet engines, helicopter rotor blades, turbines, and coal conversion plants. Considering that the abrasive particles used caused solid particle erosion by hitting moving blades, valve holes, pipe connections, pipe elbows, and other surfaces, the primary aim of the study was to reduce the erosion rate of BFR/EP composite pipes; that is, to accumulate comparison data regarding their working life [10,11,12,13,14].

The experiments were conducted using an ASTM G76-95 test set [15,16] in which erodent particles were impacted on the test sample surface with dry and compressed air, and the erosion wear condition was examined. The impact velocity was determined by the double-disc method by using the pressure difference through the valves positioned on the unit for use in the experiments [17,18,19,20]. To simulate the use of BFR/EP composite pipes and to contribute to the application of the results in the field, the impingement angle, which is caused by the erodent particles to affect the pipe outer surface, was also adjusted by rotating the sample holder around its own axis.

As a result, in this study, the samples in the form of filament winding and composite pipes used in defense industry applications had five different impingement angles (30, 45, 60, 75, and 90°), a 34 m/s impact speed, and an approximately ~600 μm abrasive particle size. To reveal the effect that will occur on the outer surface of the pipe, the solid particle erosion wear behaviors of the winding pattern structures in the lamina and zigzag regions were examined, and the erosion rate and impingement angle graphs were created. The values of the ductile and semi-ductile orientation were obtained as a result of the differences in the winding pattern structures in the axial and radial orientations in the literature studies. The microscope and SEM images of the test samples were examined, and the reasons for the experiment results’ variability were interpreted.

## 2. Materials and Method

### 2.1. Test Materials

The two basic materials used in the production of composite pipes are basalt fiber and epoxy resin; thus, monofilament basalt fibers with an average diameter of 13 µm fibers and 400 g per 1000 m fiber length were procured from Kamenny Vek Co, Dubna, Russia. Bisphenol-A epoxy resin (EPIKOTE™ 828 LVEL) and an anhydride-based curing agent (EPIKURE™ 866), which are frequently used in the filament winding method, were chosen. These main components were used as received in the production of composite pipes. The mechanical properties of the basalt fiber and epoxy resin used in the production of filament wound composite pipes are provided in Table 1.

### 2.2. Production of Composite Pipes

In this study, basalt fiber-reinforced epoxy (BFR/EP) composite pipes were produced with the filament winding method in the Izoreel Composite Insulation Materials Ltd. (Izmir, Turkey) facility in a configuration of [±55]_4_ windings on a CNC filament winding machine with filament winding parameters of a 10 N fiber tension and an average bandwidth of 11 mm. Basalt fibers were wetted by being passed through an epoxy resin bath at 80 °C for impregnation and helically wound on a 1 m-long mandrel with a wall thickness of 1.85 mm and an outer diameter of 72.2 mm, as shown in detail in Figure 1b. After the filament winding process was completed, the composite pipes were first cured at 120 °C for three hours in a horizontal rotary kiln and finally post-cured at 140 °C for three hours.

The main aim of the study was to examine the erosion wear behavior on the surfaces of the pipes in the regions shown in Figure 2 of the BFR/EP composite pipes whose production processes had been completed. Considering this basic priority, to categorize the erosion resistance in the pipes against particle impact, erosion tests in a direction parallel and perpendicular to the pipe axis, which are predicted to contribute to the significance of the inner surface deformation priority, were evaluated. However, by concluding that the effects of external surface deformations of pipes on erosion resistance are more important in industrial applications and defense industry solutions, determining the erosion rate differences that occur in the zigzag regions of the intersection points and the laminate extending along the width of the basalt fibers obtained from the outer surfaces of the pipes according to the filament winding method was prioritized.

Hardness and ring tensile strength tests were applied to determine the mechanical properties of the BFR/EP composite pipes. The hardness of the pipes was measured using the Rockwell hardness tester according to the ASTM D785 standard, and the average hardness value according to the standard L scale of the test device was determined to be 115.6 ± 4.2 HRL. The hoop tensile strength test was also applied according to the ASTM D2290 standard, and an average of 689.3 ± 46.9 MPa was obtained at a test speed of 5 mm/min. After the composite pipe production processes were terminated with curing applications and the mechanical test data were obtained, after a waiting period of at least 24 h, the pipe was cut in 50 mm lengths and weighed on a precision balance to ensure a compatibility that could simulate the industrial and usage area of the particle effect. As shown in Figure 3a, for solid particle erosion wear tests, the design structure used in the tests of the internal flow direction of the pipe, that is, of the axial orientation, was constructed. In these experiments, the impact velocities of the particles in the laminated and zigzag regions were kept constant, and the results of the solid particle erosion wear tests at the five different impingement angles shown in Figure 3b were compared and examined.

The schematic design for the tests of the radial orientation of the outer surfaces of the BFR/EP composite pipes perpendicular to the sample holder is shown in Figure 4a. The impact velocity of the particles in the laminated and zigzag regions was kept constant in these experiments, and the results of the KPE wear tests at five different impingement angles are shown in Figure 4b.

### 2.3. Test Installation

The tests were performed in an erosion wear test installation that met the ASTM G76-95 standard requirements, as shown schematically (the main components used on the unit are indicated) and in photographs in Figure 5.

After transferring the abrasive particle to the main tank for use in the experiments, the valve between the tanks was opened, and the particles were transferred to the pressurized tank. The pressure-regulating valves used in the system adjusted the pressures at the entrance of the pressurized tank and the nozzle, and the settings were controlled by manometers. The sample holder component on the test installation was used to adjust the impingement angles of the test samples of 30, 45, 60, 75, and 90°, as well as the distance between the nozzle and the test sample in a way that did not affect the impact velocity.

Particle selection is critical in determining the abrasive particle impact rate, which has an important place in solid particle erosion wear. The SEM image is presented in Figure 6. The average ~600 μm size of the Al_2_O_3_ abrasive particles (density 3.94 g/cm^3^, hardness 9 Mohs, and melting temperature 1950 °C) was used.

In the velocity measurement of particles, high-speed photography [21], a laser doppler anemometer [22], and the double-disc method [17] are velocity measurement techniques that are accepted as standard and widely used. The double-disc method is the most commonly used method for determining impact velocity values, and in the experimental study, the abrasive particle impact velocity was set at 34 m/s using this method, in which two metal discs made of phosphor bronze are connected to a common shaft that rotates beneath the nozzle, as shown in Figure 7a. These discs are connected to the drive motor, which precisely determines the rotation of the discs. After the interaction of the drive motor movement and the particles directed from the nozzle, the erosion wear marks formed on the lower disc by particles passing through the slit in the upper disc are shown in Figure 7b.

The distance between the discs (L) and the distance of the nozzle to the discs were defined at the start of the double-disc method and were determined based on the distance between the nozzle tip and the test sample. The drive motor could control the rotation speed (v) of the sample and the discs. At the end of the velocity measurement experiments of three replicates, the distance between the erosion marks formed on the lower disc (S) and the radius of the marks to the disc center (r) were measured and written in Equation (1), and the abrasive particle velocity (υ) was theoretically calculated [17].
υ = (2 × π × r × ν × L)/S(1)

To investigate the effects of the abrasive particle mass (Q_p_) acting on the test sample on the erosion rate, cumulative erosion tests were performed. The wear conditions of the samples were observed in these experiments, and the erosion wear of the composite pipes was investigated by measuring the weight of the test sample with a precision balance with ±0.0001 g sensitivity after each test. As the result of three repeated tests, it was decided to use 2 kg of abrasive particles in the experiments conducted at a 34 m/s impact speed of the abrasive particles on the outer surfaces of the pipes and at room temperature, and the erosion rate value was determined using Equation (2):ER = ΔW/Q_p_
(2)

The erosion rate determined by this formulation was expressed as ER and calculated in mg/kg. The mass loss (ΔW) of BFR/EP materials occurring in the test samples measured with precision balance before and after the experiment was represented in mg units. The mass of abrasive particles hitting the outer surfaces of the pipes was given as Q_p_, and erosion rates (kg) were obtained.

## 3. Results and Discussion

The BFR/EP composite pipes produced in a [±55]_4_ winding configuration with fiber tension and bandwidth variability were under the impact of a 34 m/s impact velocity of angular geometrical abrasive particles with an average diameter of 600 μm. At impingement angles of 30, 45, 60, 75, and 90°, solid particles were subjected to erosion wear. In the axial and radial directions, the erosion resistance of the winding pattern structures in the lamina (±55 angle-ply laminate region) and zigzag (±55 zigzag region) regions of each composite pipe outer surface, determined as the test sample, was compared.

### 3.1. Effect of Impingement Angle on Erosion Rate

The graph in Figure 8 depicts the changes in impingement angle and erosion rate in the lamina and zigzag erosion wear test positions, where the effect on the pipe’s outer surface in the axial direction was investigated. When the related graph is examined in depth, it may be concluded that while the lamina and zigzag forms exhibit similar tendencies, the impingement angle causes a difference in wear due to the differentiation of abrasive particles striking in parallel and perpendicular to the fibers due to the winding configuration [±55]_4_ feature. After the 30° impingement angle, it was determined that the erosion rate decreased in line with the increase in the angle between the nozzle and the pipes from oblique to vertical impact for both test positions. In particular, the 30° and 45° data exhibited close erosion resistance and are compatible with the literature; that is, a situation similar to ductile-prone erosional wear has occurred [12,23]. In other words, in these tests, the scraping effect of oblique impact was dominant, and separation from the surface was accelerated.

The graph in Figure 9, on the other hand, contributes to the comparison of the solid particle erosion resistance of the outer surface of the pipe as a result of the radial action of the erosive particles, examines the effects of the lamina form results and the zigzag form in flat and opposite positions, and shows the changes between the impingement angle and the erosion rate. Although the erosion wear results in this direction are compatible with similar effects in the literature [20], it was observed that the formation of the wearing zone changed in the radial direction tests in contrast to the axial direction tests. It should be noted that a sample application of pipe form erosion wear tests had not been carried out until the current study. The deformation caused by the abrasive particles on the pipe surface reached its maximum level in these experiments at a 60° impingement angle. In other words, erosion resistance decreases after a 30° impingement angle up to a 45° and even a 60° impingement angle, and then significantly increases at 75° and 90° impingement angles, and the wear course of semi-ductile materials has been encountered in the literature [24,25].

The incubation event on the pipe surface of the abrasive particles hitting the material surface at right angles when the oblique impact form tends towards the vertical impact was also interpreted as a reduction in erosion rate with the effect of micro-scale deformation and hardening due to the incubation event on the pipe surface of the abrasive particles hitting the material surface at right angles when the oblique impact form tends towards the vertical impact. Variations in the orientation of the composite pipes can make a significant positive contribution to the problem caused by abrasive particles on the surfaces, according to these experiments.

In the experiments, the impact velocity of the particles hitting the outer surfaces of the composite pipe and the abrasive particle size and type were not questioned, and the focus was placed on the extent to which the impingement angle variability and exposure to particle contact from the axial and radial directions could cause changes in the erosion rate.

Figure 10 shows a graphical representation of the changes in impingement angle and erosion rate for all composite pipe orientations whose wear conditions were questioned within the scope of this experimental study overall, with the effect of the combined comparisons on the significance highlighted. This graph shows, in particular, how important the impact of the test position on the impingement angle is. A strong wear resistance was observed in all test positions when conducted at a 90° impingement angle. When only axial tests were examined, it was discovered that the zigzag region had higher erosion resistance than the lamina region at the 30° impingement angle where the maximum wear occurred, and the erosion rate was reduced by 10%. When only the radial direction tests were examined, it was discovered that the erosion rate in the zigzag regions decreased by 30% for the 60° impingement angle, which is where the most wear occurred, when tests were conducted in both radial and axial directions; however, it was clear that the zigzag region had the strongest tendency to resist wear. When the maximum erosion rate occurred in the lamina region in radial direction tests and in the lamina region in axial direction tests, the wear resistance increased by 30% in the axial direction tests with the same particle impact velocity and abrasive particle mass.

As can be seen in the design in which the radial flow is defined in Figure 4, as the impingement angle decreases, the particle flow on the pipe surface changes due to the angular position between the sample and the nozzle, and the damaged area on the pipe’s surface also changes. The effect of the cosine component of the impingement angle increases at 60° compared to 75° and 90°. The regional contact of the particles on the pipe’s surface is in a larger area at 45° and 30°. The ploughing effect of the cosine component of erosive particles deeply damaged the pipe’s surface, causing larger wear. Looking at the stereo microscope images in Figure 11, Figure 12, Figure 13 and Figure 14, it has been shown that the eroded areas vary depending on the impingement angle. The effect of the cosine component of the impingement angle increases at 60° compared to 75° and 90°. The regional contact of the particles on the pipe’s surface is in a larger area at 45° and 30°. As a result, the maximum wear occurred at 60°, depending on the impact angle and pipe surface position.

### 3.2. Effect of Impingement Angle on Erosion Efficiency

When what effect the impingement angle has on the efficiency of solid particle erosion is questioned, it is possible to obtain information about whether the materials tested are ductile or brittle in the literature [26,27]. In this context, the erosion efficiency interpretation was performed using the following formula. In Equation (3), E represents the erosion rate (mg/kg), H represents the sample hardness (Pa), ρ represents the sample density (kg/m^3^), and ν represents the impact velocity (m/s) [28,29]:η = (2 × E × H)/(ρ × ν^2^) (3)

While zero erosion efficiency is used in the realization of fragment separation without surface breakage, the 0–1 range in erosion efficiency represents the breakage situation caused by abrasive deformation. Furthermore, the initiation of lateral or radial cracks, which causes erosion efficiency to exceed 100%, and the separation of large pieces of material from the surface revealed a significant result.

The changes in the efficiency values of 30, 45, 60, 75, and 90° impingement angles, respectively, were calculated using Formula (3), and the graph in Figure 15 was created. While the angle-specific comparison effect determines priority dominance, radial and axial pipe outer surface orientations have a direct impact on erosion resistance. Furthermore, the findings revealed that the separation from the sample surface was interpreted as the abrasive particle activity gradually penetrating the surfaces. In addition, it was discovered that the outer surface orientations of filament winding composite pipes had a positive effect on erosion efficiency and an inverse effect on erosion resistance.

### 3.3. Macro- and Micro-Scopic Studies of Worn Surfaces

SEM images of the microstructures of the epoxy matrix composite pipes produced with basalt fiber additives in CNC filament winding machines were obtained, covering the general and regional areas that were the particular focus of this study. These photographs were obtained to help determine the deformations in the basalt fiber-specific matrix of the axial and radial lamina and zigzag test positions at 30°, 60°, and 90° impingement angles and the differences between them. Moreover, stereomicroscope images of the worn surface of the BFR/EP composite pipe in the lamina and zigzag regions (in ±55 lamina and ±55 zigzag regions) in the axial and radial positions were obtained after wear. To define the transition of general and local wear, SEM and stereomicroscope images were combined to present the superficial deformation in an integrated manner to compare the fiber and/or matrix damage dominance. The effect of change on the surfaces of the axial orientation lamina and zigzag test position at a 30° impingement angle is shown in Figure 11a,b, and it was concluded that the depth effect and regional width potential of the traces formed on the sample surfaces were similar. In the case of materials with a ductile tendency, the condition of the separated layers was realized when the compatibility of the erosion rate obtained from the experiments and the consistency of the surface material deformation were also questioned. In Figure 11, both matrix and fiber breakages have been seen in the eroded region. Additionally, we observed the ploughing mechanism in areas where the matrix and fiber were damaged. In Figure 11b, we found intensive fiber breakages. When we look at the erosion rate seen in Figure 8, we can observe that the ploughing mechanism plays a role in increasing the wear.

In Figure 12a,b, the experiments were carried out in the axially oriented test position. However, when the main difference is the impingement angle (90°), it is understood that in the case of variability from the oblique impingement angle to the perpendicular impingement angle, the effect of the main difference observed in the surface form, especially in the abrasive particle ploughing role, turns into pressing the surface inward; that is, the embedding situation in a way that coincides with the existence of incubation. As a result, fiber breakages were seen at the 30° impingement angle, and partial matrix damages have been observed at the impingement angle of 90°. This showed that the erosion rate resulted in a decrease.

After observing the effects of the axial orientation on the surface changes of both the lamina and zigzag test positions and the minimum (90°) and maximum (30°) erosion wear values at the impingement angle on the surfaces, comparisons of SEM images were performed when the angle change of the erosion wear differences of the radial orientation was prioritized. Figure 13a–c show the maximum erosion rate at a 60° impingement angle, confirming the finding that having more than one layer on the lamina, as well as zigzag opposite and zigzag straight surfaces, results in a much more erosive effect.

In addition, for the radial orientation lamina test position, it was determined that the highest erosion rate was reached as a result of the variability in the resultant force distribution in the Cosine components due to the winding feature at a 60° impingement angle, [±55]_4_. The point that is especially determined in the separation of the radial and axial orientation of the filament winding BFR/EP composite pipes is to examine the surfaces of the samples where the highest erosion wear is observed, and the consistency error is prevented in the comparisons.

As shown in the image of the stereomicroscope in Figure 13a, deeper damage occurred. In the region shown here, the matrix was ploughed, and the fibers were broken. In Figure 13b,c, the ploughing effect could not reach the depth, but the ploughing mechanism and fiber breaks occurred. Wear increased in the lamina area where deeper damage took place. In the zigzag regions, as shown in Figure 2, it was observed that the fibers supported each other and increased the erosion resistance.

In the SEM image comparisons in Figure 14a,b, the impingement angle difference of the radial orientation test position was examined. Although the axial orientation compatibility in these images was similar, the impingement angle at which maximum erosion wear occurred was 60°, resulting in a material tendency transition to semi-ductile. The rolling effect on the sample surface was prioritized by the particles in the right-angle application experiments.

It was observed that the matrix of the composite pipe was ploughed more strongly at the impingement angle of 60° and fiber damage occurred in a larger area. At the 90° impingement angle, the matrix was intense in the worn area and partial fiber damage was observed.

Microscope images showed that significant differences were found in the axial and radial orientation data of the particle impact case of the basic test pair, which reinforces its connection with the literature on BFR/EP composite pipes obtained as a result of filament winding and changes the erosion wear effect between them depending on the test position. With the publication of this finding, it is suggested that external surface orientation data derived from production will aid in the prevention of wear in applications where the pipe’s usage options are numerous.

## 4. Conclusions

In this experimental study in which the effect of solid particle erosion wear on the outer surface of the pipe was questioned specifically for the test position, the following results were obtained.

The resistance of BFR/EP composite pipes with a [55]_4_ filament winding configuration to particle contact was examined specifically for pipe application for the first time in the literature.

In the case of pipe outer surface axial positioning, regardless of lamina and zigzag form differences, the maximum erosion rate was observed in all pipe samples at a 30° impingement angle, a trend similar to that seen in ductile materials. The erosion rate values, on the other hand, decreased in tandem with the increase in the impingement angle. This was because the lamina position resulted in a higher erosion rate, and the abrasives hitting parallel to the fiber orientation aided in the removal of parts from the outer surface.

Unlike the axial position, the 60° impingement angle had the lowest level of erosion resistance in the erosional wear of the pipe outer surface in the radial axial position. When the abrasive contact parallel to the basalt fibers was dominant and the situation shared in the production images was examined, it was determined that the orientation of the fibers caused an increase in the rate of erosion thanks to the arrangement parallel to the matrix, instead of acting as a barricade to the particles.

Through the [±55]_4_ filament winding configuration, the impact effect of the particles on the samples significantly increased due to the normal and tangential components in the variation of the particle impingement angle between 30° and 90°, and this situation directly affected the deformation on the outer surface of the pipe.

As a result of the basalt fibers not resisting cracking in the matrix with the change of the test position, SEM images of the worn pipe surfaces revealed a wear mechanism that favored the breaking of fiber/matrix bonds.

## Figures and Tables

**Figure 1 polymers-15-00319-f001:**
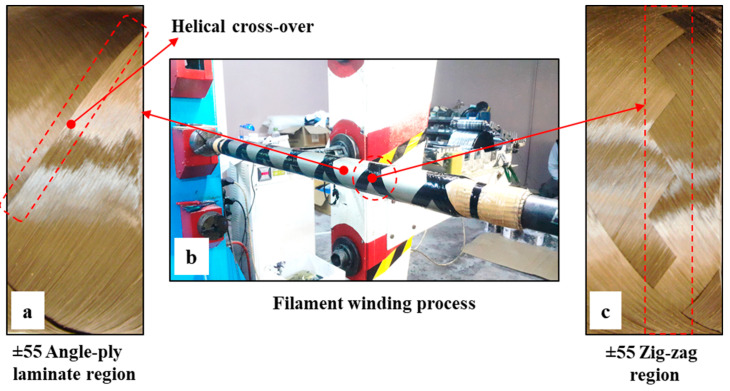
BFR/EP filament winding composite pipes: (**a**) ±55 angle-ply laminate region, (**b**) filament winding process, and (**c**) ±55 zigzag region.

**Figure 2 polymers-15-00319-f002:**
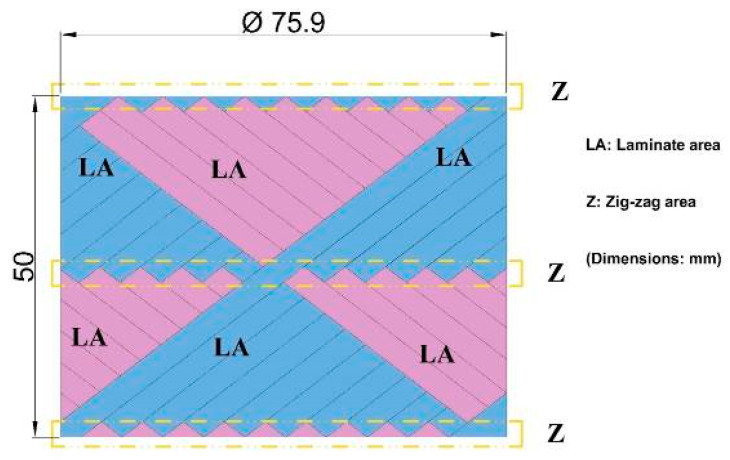
Filament winding variations on the outer surfaces of BFR/EP composite pipes.

**Figure 3 polymers-15-00319-f003:**
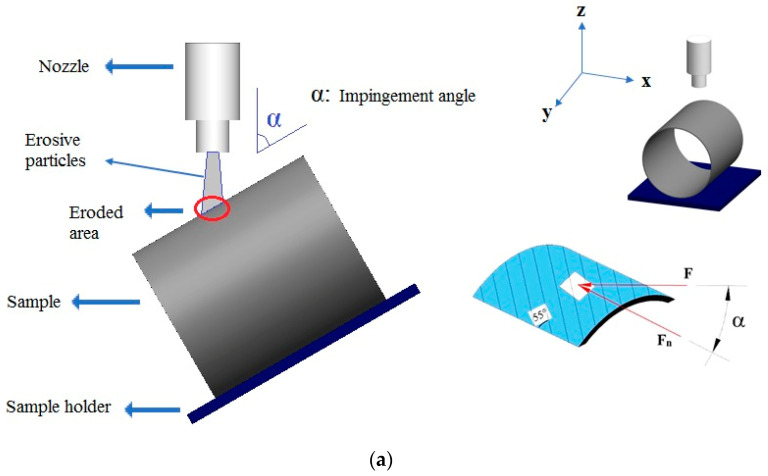
Pipe outer surface axial positioning test details: (**a**) SPE wear test design, and (**b**) schematic views of the variation of the impingement angle in the experiments (F: impact force, F_n_: normal force on laminate).

**Figure 4 polymers-15-00319-f004:**
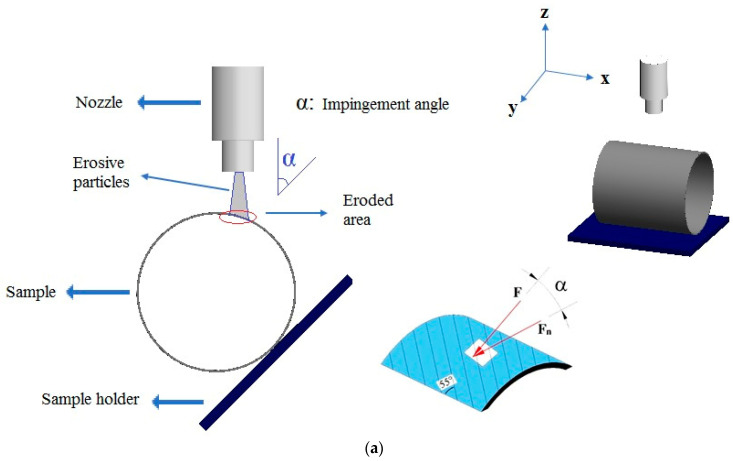
Pipe outer surface radial positioning test details: (**a**) SPE wear test design, and (**b**) schematic views of the variation of the impingement angle in the experiments (F: impact force, F_n_: normal force on laminate).

**Figure 5 polymers-15-00319-f005:**
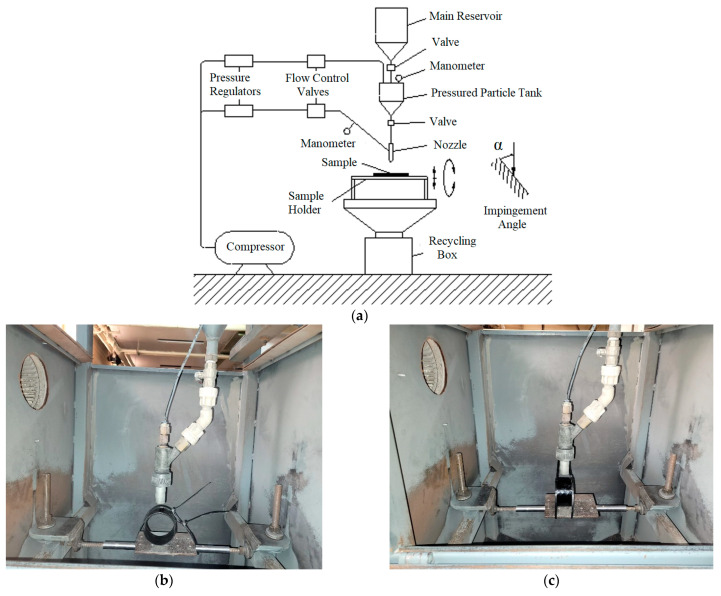
Experimental setup: (**a**) detailed schematic representation of components, (**b**) pipe axial positioning, and (**c**) pipe radial positioning.

**Figure 6 polymers-15-00319-f006:**
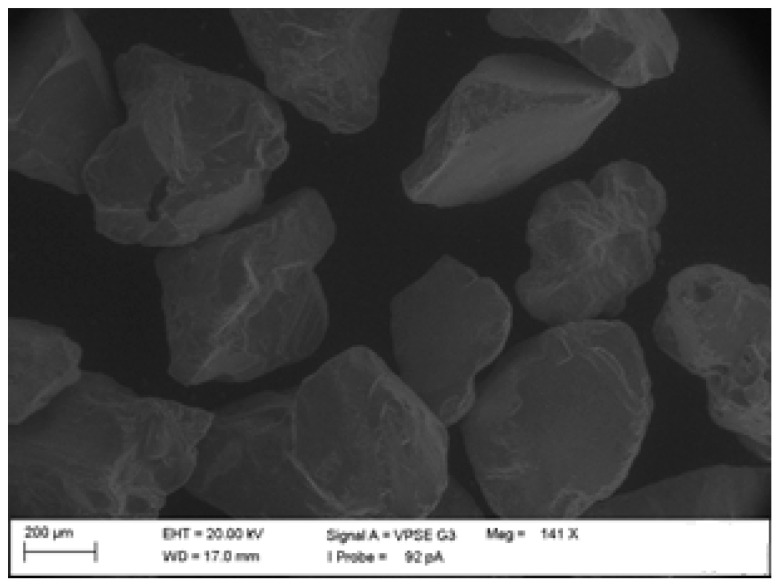
SEM image of Al_2_O_3_ abrasive particles with angular geometry.

**Figure 7 polymers-15-00319-f007:**
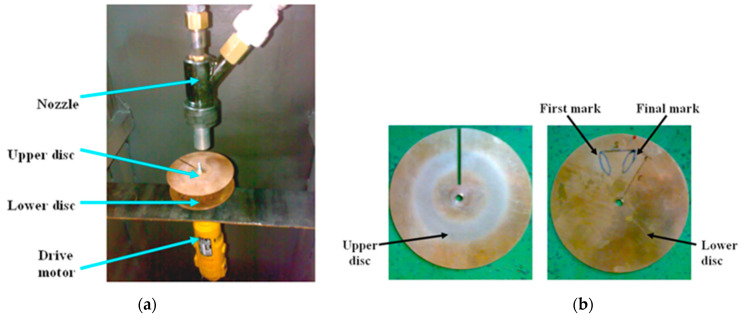
Double-disc method: (**a**) position of motor, nozzle, and discs, and (**b**) erosion marks on discs.

**Figure 8 polymers-15-00319-f008:**
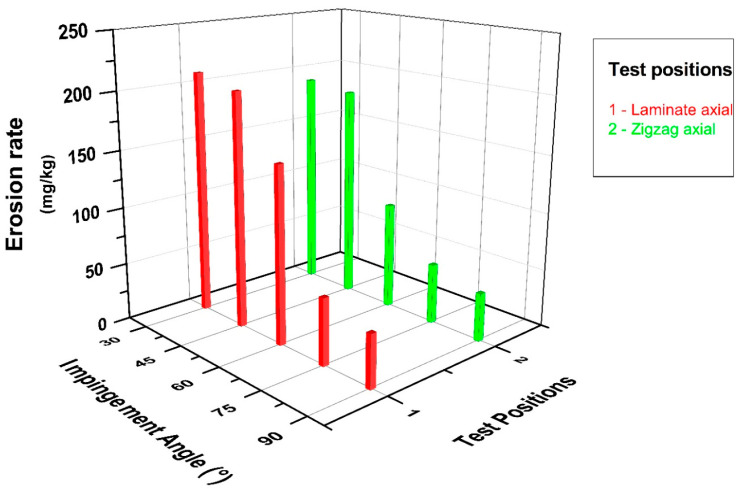
BFR/EP composite pipe outer surface axial orientation erosion rate–impingement angle relationship.

**Figure 9 polymers-15-00319-f009:**
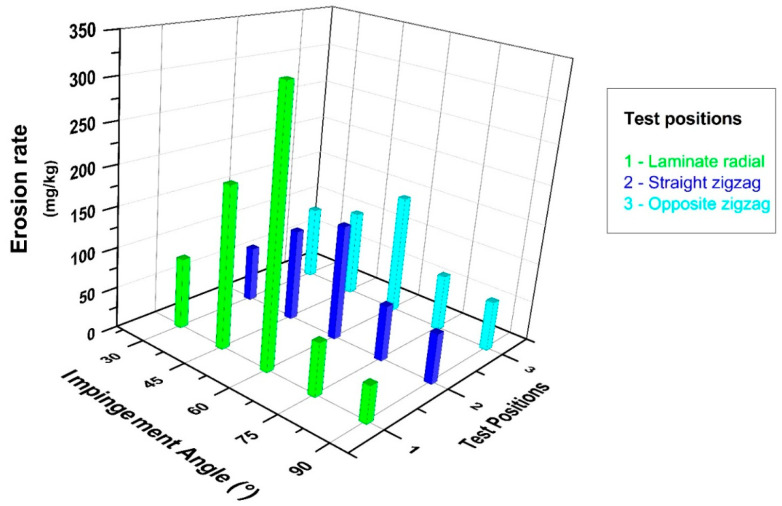
BFR/EP composite pipe outer surface radial orientation erosion rate–impingement angle relationship.

**Figure 10 polymers-15-00319-f010:**
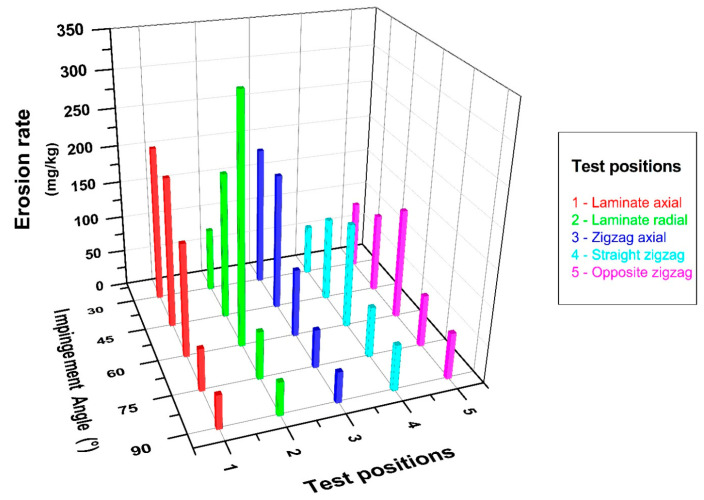
BFR/EP composite pipe outer surface radial and axial orientation erosion rate–impingement angle relationship.

**Figure 11 polymers-15-00319-f011:**
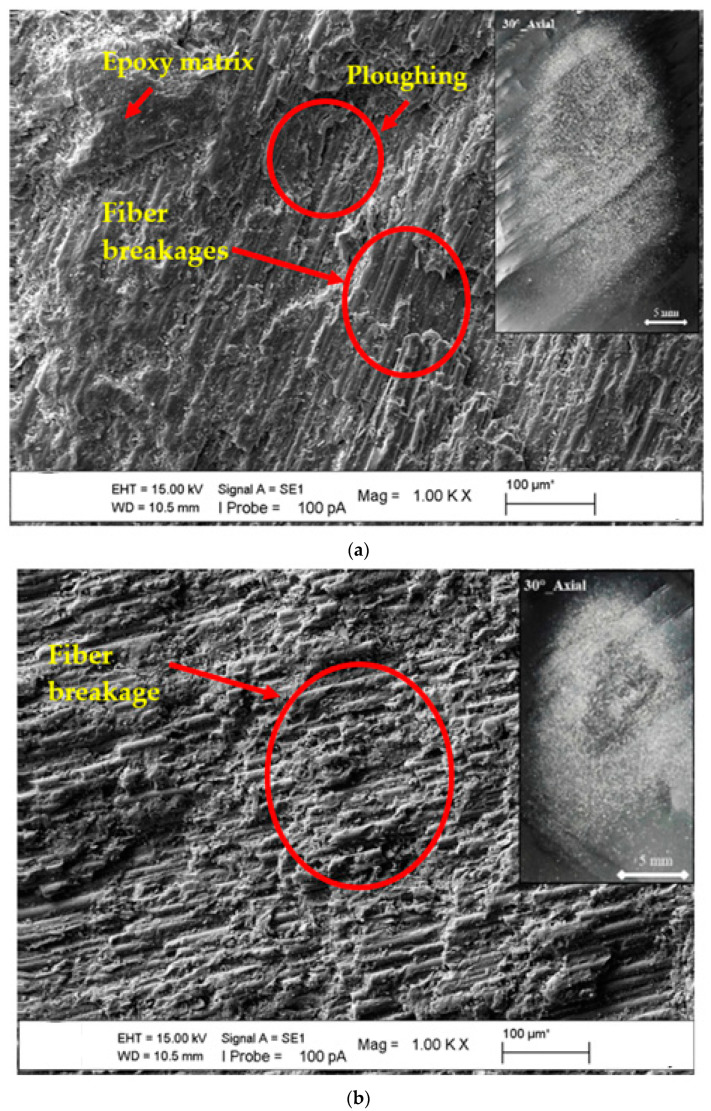
SEM images of matrix damages and fiber breakage on the surface at the 30° impingement angle with axial orientation: (**a**) lamina region and (**b**) zigzag region.

**Figure 12 polymers-15-00319-f012:**
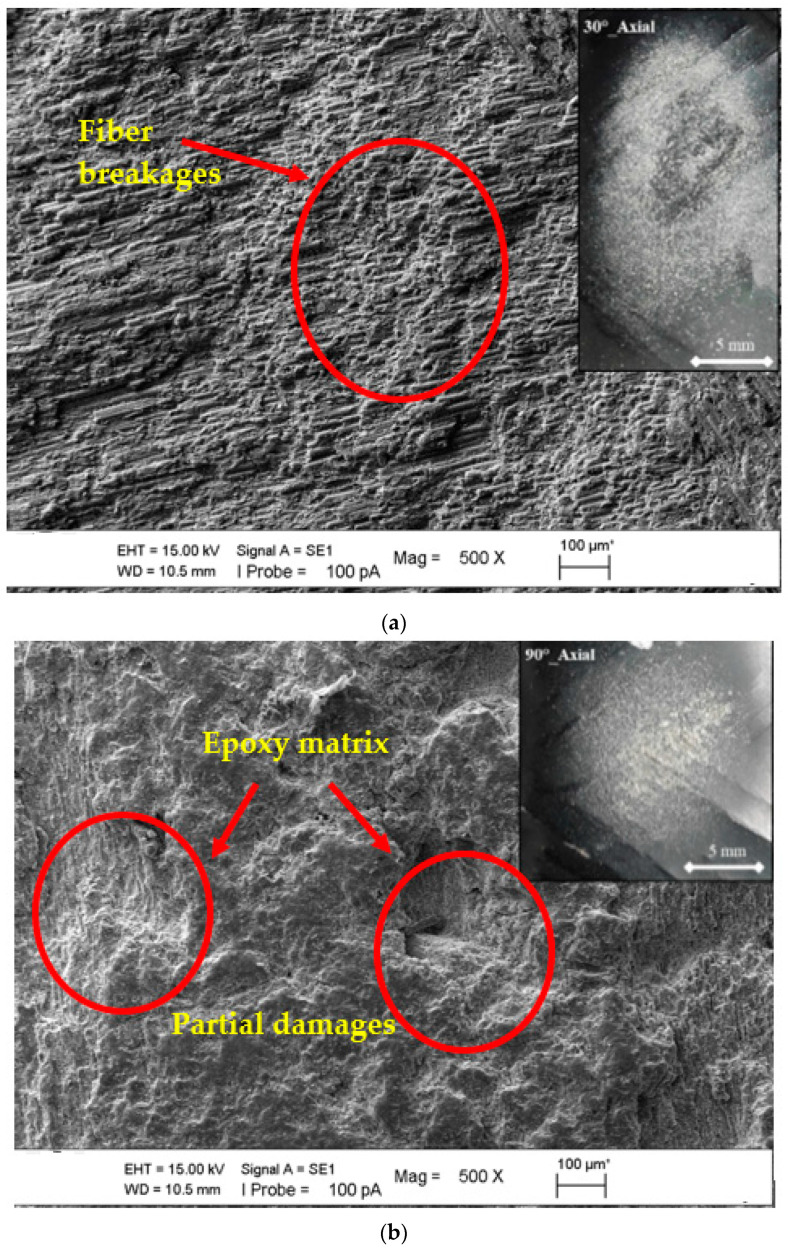
SEM images of matrix damages and fiber breakage on the surface at the axial orientation with the changing impingement angle: (**a**) 30° and (**b**) 90°.

**Figure 13 polymers-15-00319-f013:**
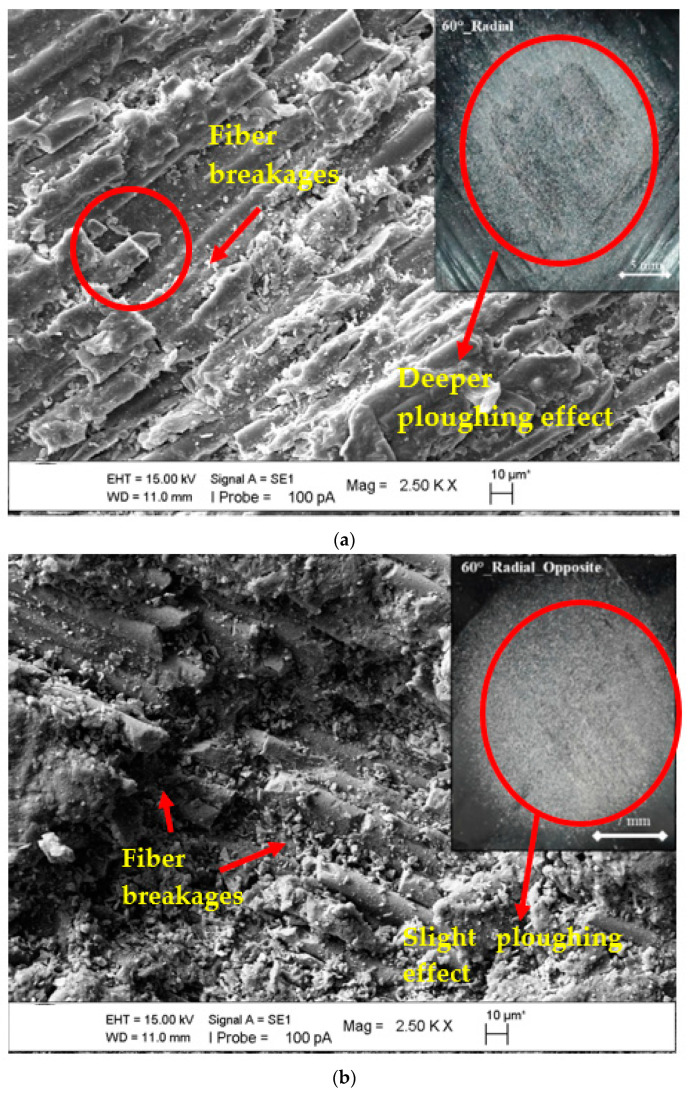
SEM images of matrix damages and fiber breakage on the surface at the 60° impingement angle with radial orientation: (**a**) lamina region, (**b**) zigzag opposite, and (**c**) zigzag straight.

**Figure 14 polymers-15-00319-f014:**
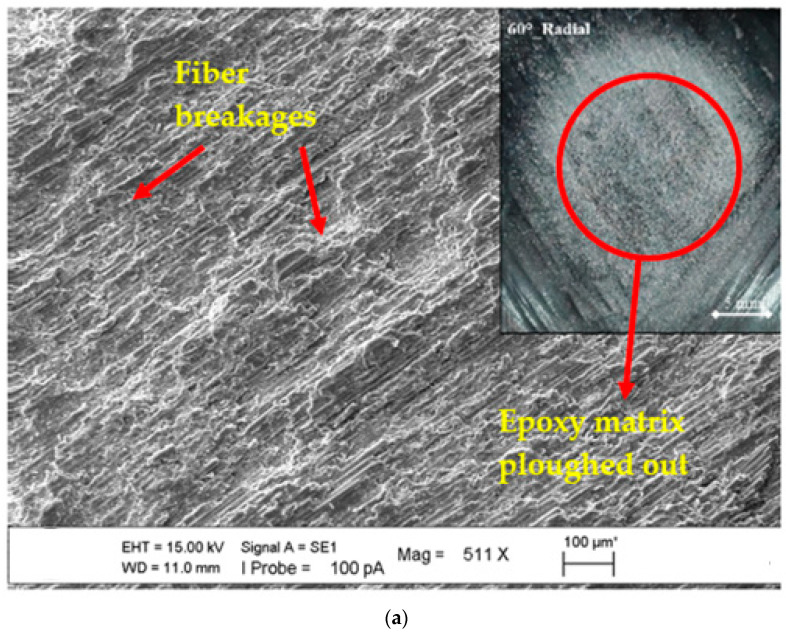
SEM images of matrix damages and fiber breakage on the surface at the radial orientation in the lamina region with the changing impingement angle: (**a**) 60° and (**b**) 90°.

**Figure 15 polymers-15-00319-f015:**
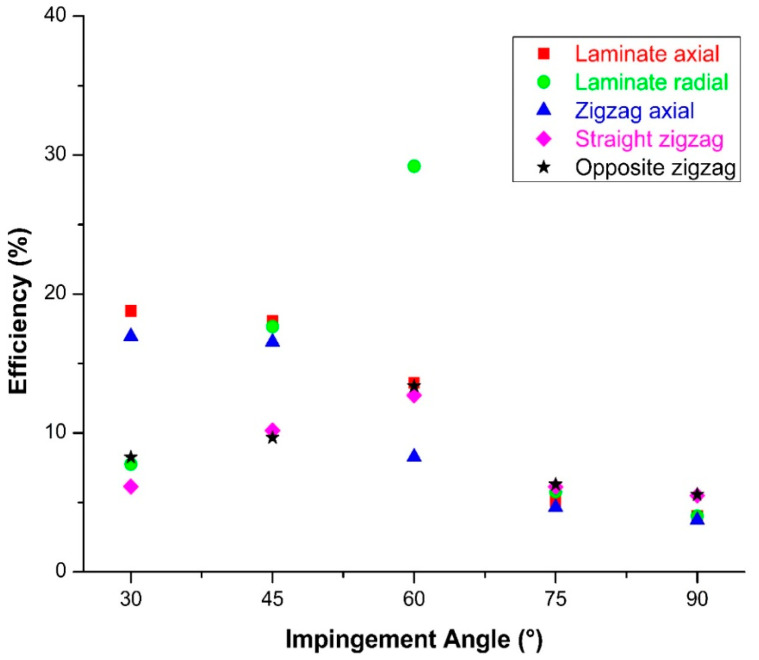
Effect of impingement angle on erosion efficiency in BFR/EP composite pipe outer surface radial and axial orientations.

**Table 1 polymers-15-00319-t001:** Properties of resin and fiber used in the production of BFR/EP filament winding composite pipe.

Type	Elasticity Modulus E, GPa	Ultimate Tensile Strength σ, MPa	Strain at the Break ε, mm/mm	Density ρ, g/cm^3^
Basalt Fiber	90–95	2900–3200	-	2.48
Epoxy Resin	3.2	70–75	4–5	1.25

## Data Availability

The data presented in this study are available on request from the corresponding author.

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
