# Peer review of "Solid Particle Erosion Behavior on the Outer Surface of Basalt/Epoxy Composite Pipes Produced by the Filament Winding Technique"

_polymers, 2023, doi:10.3390/polym15020319_

Round 1

Reviewer 1 Report (Previous Reviewer 1)

The authors have addressed all my concerns and comments. The paper is in a good shape to be published.

Reviewer 2 Report (Previous Reviewer 3)

The Manuscript is accepting in the present form

This manuscript is a resubmission of an earlier submission. The following is a list of the peer review reports and author responses from that submission.

Round 1

Reviewer 1 Report

In this paper, the authors have carried out wear experiments on basalt-epoxy composite material. They carried out the erosion test at various impingement angles and observed the erosion ratio and later characterized it with SEM images.

While the results obtained from these experiments are interesting and will be useful to the readers of the Polymers journal, the paper lacks depth of discussion around the results. The authors have done the experiments and reported their findings but haven’t done a detailed investigation as to why they observe such results. I can recommend the publication of the paper only after some major modifications to the paper.

Reviewer 2 Report

In the manuscript, abrasive impact tests were carried out on epoxy (EP) composite pipes with  [±55]4 winding configuration were produced on CNC filament winding machines (10 N fiber tension and ~11 mm bandwidth) with the addition of a matrix by two different winding methods, a 34 m/s impact speed was set using the double disc method, and five different abrasive particle impact angles were used to determine the erosive effect on the outer surfaces of filament wound composite pipes under the influence of 600 m abrasive particles with angular geometry in the test set complying with the ASTM G76-95 standard. The results of repeated experiments at various angles (30°, 45°, 60°, 75°, and 90°) were interpreted. The winding patterns in the lamina (±55 angle-ply laminate region) and zig-zag (±55 zig-zag region) regions of BFR composite pipes were determined to have significant effects on solid particle erosion resistance as evidenced by the SEM images. It provides the necessary theoretical and experimental basis for the optimal design of epoxy (EP) composite pipe structure and the selection of working conditions, but the following problems need to be explained and explained by the author.

1. In the test, the author used ASTM G76-95 standard test device to carry out abrasive impact test on the outside of epoxy (EP) composite pipe around the structure, and the impact speed reached 34 m/s in the test, which may appear in the working position simulated by the author such as blade, valve hole, pipe joint, pipe elbow and so on. However, it should be noted that most of the abrasive motion and wear behaviors at this speed occur inside the tube, and the opposite curvature will cause changes in the motion characteristics of gas-solid two-phase flow and the contact characteristics between abrasive particles and tube wall materials. Therefore, the reviewer believes that the experimental design in this manuscript is unreasonable. Abrasive impact tests should be performed on the inner wall of epoxy (EP) composite pipe with wound structure.

2. In the manuscript, scanning electron micrograph analysis was carried out on erosion rate and erosion damage on the surface of epoxy (EP) composite pipe. The microstructure of erosion position shown in scanning electron micrograph showed that the basalt fiber in the tube wall composite was fractured under the high-speed impact of abrasive particles. However, the specific mechanism and action mechanism of the influence of structural design on wear resistance are not strongly explained in the pictures and descriptions.

In view of the above, the reviewers consider that the manuscript is not yet ready for publication under the circumstances.

Reviewer 3 Report

- Abstract needs to summarize

-Manuscript has some typos, revise carefully and correct it

- Line 28 changes “(30°, 45°, 60°, 75°, and 90°)” to “(30, 45, 60, 75, and 90°)”

- 2. Materials and method, include information for materials purifications or include the sentence “used as received”

- Figures quality needs to improve at a minimal 300 dpi

- Manuscript needs to revise by native English speaking

- Lines 173 and 219, etc. change “30°, 45°, 60°, 75°, and 90°” to “30, 45, 60, 75, and 90°”

- Manuscript has 15 Figures with 27 images, summarize Figures and use supplementary materials for no important images

- Manuscript has some interesting results but needs to improve the discussion for all Figures

- Manuscript has only one reference from 2022, include more recent references